# Community-based rehabilitation training after stroke: results of a pilot randomised controlled trial (ReTrain) investigating acceptability and feasibility

Sarah G Dean,[1] Leon Poltawski,[1] Anne Forster,[2] Rod S Taylor,[1] Anne Spencer,[1] Martin James,[1,3,8] Rhoda Allison,[4] Shirley Stevens,[1] Meriel Norris,[5] Anthony I Shepherd,[6] Paolo Landa,[1] Richard M Pulsford,[7] Laura Hollands,[1] Raff Calitri[1]

For numbered affiliations see end of article.

**Correspondence to**
Sarah G Dean;
s.dean@exeter.ac.uk

## ABSTRACT

**Objectives** To assess acceptability and feasibility of trial processes and the Rehabilitation Training (ReTrain) intervention including an assessment of intervention fidelity.

**Design** A two-group, assessor-blinded, randomised controlled trial with parallel mixed methods process and economic evaluations.

**Setting** Community settings across two sites in Devon.

**Participants** Eligible participants were: 18 years old or over, with a diagnosis of stroke and with self-reported mobility issues, no contraindications to physical activity, discharged from National Health Service or any other formal rehabilitation programme at least 1 month before, willing to be randomised to either control or ReTrain and attend the training venue, possessing cognitive capacity and communication ability sufficient to participate. Participants were individually randomised (1:1) via a computer-generated randomisation sequence minimised for time since stroke and level of functional disability. Only outcome assessors independent of the research team were blinded to group allocation.

**Interventions** ReTrain comprised (1) an introductory one-to-one session; (2) ten, twice-weekly group classes with up to two trainers and eight clients; (3) a closing one-to-one session, followed by three drop-in sessions over the subsequent 3 months. Participants received a bespoke home-based training programme. All participants received treatment as usual. The control group received an exercise after stroke advice booklet.

**Outcome measures** Candidate primary outcomes included functional mobility and physical activity.

**Results** Forty-five participants were randomised (ReTrain=23; Control=22); data were available from 40 participants at 6 months of follow-up (ReTrain=21; Control=19) and 41 at 9 months of follow-up (ReTrain=21; Control=20). We demonstrated ability to recruit and retain participants. Participants were not burdened by the requirements of the study. We were able to calculate sample estimates for candidate primary outcomes and test procedures for process and health economic evaluations.

### Strengths and limitations of this study

► A community-based exercise intervention after stroke developed with service users.
► A pilot randomised trial developed following Medical Research Council guidelines on complex interventions.
► Mixed-method approach for answering feasibility and acceptability objectives.
► Objective physical activity capture via accelerometry but subjective self-report outcomes for functional mobility and psychosocial measurements.

**Conclusions** All objectives were fulfilled and indicated that a definitive trial of ReTrain is feasible and acceptable.

**Trial registration number** NCT02429180; Results.

## INTRODUCTION

Five years after initial stroke, one in three individuals have residual physical impairment,[1] equating to over 300 000 individuals in the UK living with disability from stroke.[2] Provision of stroke rehabilitation is typically front loaded, with resources focused on in-patient care and early supported discharge. Support tapers off after a few months,[3] with many individuals reporting unmet long-term needs[4]

The National clinical guideline for stroke advise for secondary prevention that stroke survivors engage in 150 min of physical activity a week, in bouts of 10 min or more, starting light and developing across time to moderate levels of intensity.[5] However, many stroke survivors do not meet these recommendations[6 7] due to combinations of personal (eg, physical or psychological

impairments) and environmental factors (eg, lack of programmes and facilities). To address this problem, community-based programmes are promoted.[8–10] These tend to focus on cardiovascular fitness with less emphasis on functional improvements or on promoting on-going exercise self-management. National stroke guidelines[5] identify the importance of interventions for functional improvement[11] and self-management,[12] but evidence is lacking regarding these types of intervention.[13]

Action for Rehabilitation following Neurological Injury (ARNI) is an approach aimed at improving function and facilitating self-management[14] and has a detailed self-help book. The ARNI approach embodies a set of principles (eg, instilling a commitment to regular exercise) and techniques tailored to individual need. The ARNI Institute trains registered exercise professionals to deliver key ARNI techniques. Clinical Commissioning Groups (CCGs), charitable and local authorities have started to provide community-based ARNI training for stroke survivors, which has been positively received by participants, carers and practitioners[15]; however, there is currently no randomised controlled trial (RCT) evidence for evaluating its impact on stroke outcomes or its cost-effectiveness.

### Background and objectives

Using the Medical Research Council's framework for the development and evaluation of complex interventions[16] and considerable Patient and Public Involvement, we have designed a testable programme called Rehabilitation Training (ReTrain).[17–20] ReTrain is a community-based, manualised group programme combining ARNI principles and key techniques with best practice guidelines for stroke.[9 17] The overall aim of our pilot RCT was to inform the design and delivery of a definitive RCT. Our objectives were to (1) assess feasibility and acceptability of recruitment (target n=48), randomisation, allocation concealment and outcome assessment blinding; (2) determine retention rates (target of <20% attrition); (3) check ReTrain's acceptability and feasibility for participants, and refine the trainer manual; (4) test candidate outcome measures, assess their burden, levels of completion and estimate outcome variance (to inform definitive trial sample size); (5) perform process evaluation including intervention fidelity assessment and (6) calculate ReTrain costs and assess feasibility of collecting health and social service resource use.

### METHODS

A brief methods overview is provided in accordance with guidance for reporting pilot trials[21]; further details are available in the published protocol.[22] Ethics review was conducted by National Research Ethics Service Committee South West Cornwall & Plymouth (REC Ref: 15/SW/04).

### Trial design

ReTrain was a two-group, assessor-blinded, randomised controlled external pilot trial with parallel mixed methods process and economic evaluations. Eligible participants were individually randomised 1:1 to intervention (ReTrain) or control (exercise advice booklet[23]).

### Participants

Inclusion criteria were as follows: (1) diagnosis of stroke; (2) any time since stroke but at least 1 month since discharge from National Health Service (NHS) physical rehabilitation services; (3) able to walk independently indoors with or without mobility aids, but with self-reported difficulty with stairs, slopes or uneven surfaces; (4) willingness to be randomised and attend the training venue and (5) cognitive capacity and communication ability sufficient to participate.

Exclusion criteria were less than 18 years old, currently (or within 1 month of) receiving ARNI training or have contraindications to moderate to vigorous physical activity (adapted from American College of Sports Medicine guidelines[24]). Participants were recruited from two CCGs. Participants were identified by: (1) clinicians in NHS primary care, hospital and community stroke services; (2) contacts in the local Clinical Research Network and Clinical Research Facility; (3) promotion via local stroke support networks (eg, Stroke Association) and (4) word of mouth, study flyers and adverts.

### Intervention

ReTrain aims to: (1) enhance function through task-related practice, teaching compensatory techniques and providing targeted strength training (cardiovascular fitness gains also occur through these activities); (2) develop self-management skills for on-going rehabilitation; (3) deliver personalised training using negotiated goals and (4) instil a commitment to regular exercise for health improvement and longer-term maintenance. ReTrain facilitates safe and efficient practice of walking in varied terrains, kerbs, cambers and in crowds, turning and moving quickly, climbing steps and stairs without rails, getting to and from the floor without furniture or other aids and moving without mobility aids or while carrying loads. Training is based on a manual and led by personal trainers on the UK Register of Exercise Professionals (level 3 or above) who are ARNI-trained and accredited and have had additional training in the delivery of ReTrain. There was a maximum ratio of one trainer to four stroke survivors. ReTrain was delivered in a community setting (one gym, two church halls and one community centre) with twice-weekly 2-hour sessions over 3 months, comprising: an introductory one-to-one session (home visit); 10, twice-weekly group classes with up to 2 trainers and 8 clients (training venue); a closing one-to-one session (home visit); followed by 3 (one per month) drop-in

sessions. Participants completed bespoke home-based training (homework) throughout.

## Control

All participants received treatment as usual. This ranged from zero treatment to engagement with any health service(s). We requested that all trial participants did not participate in additional physical rehabilitation (either NHS or private) but we could not prevent them from doing so. We did not monitor control group participation in any treatments during the trial but did record health service use at the end of the trial for all participants. The control group also received an advice booklet about exercise after stroke.[23]

## Outcomes

*Feasibility, acceptability and process outcome:* numbers and details of those approached; recruitment and retention figures.

*Acceptability of randomisation, outcome measurement burden and the intervention:* completion of questionnaires and objective assessments, interviews with 10 intervention and 10 control group members and the trainers. *Safety*: Adverse events[25] identified via trainer and ReTrain participants (during the programme) and participant reports (all participants during 6-month and 9-month assessments).

*Intervention fidelity:* attendance registers, accelerometry, exercise 'homework' diaries, trainer completed session checklists and video analysis of (early, middle and late programme) training sessions.

We tested a range of candidate primary and secondary outcome measures.

*Primary outcomes*: Rivermead Mobility Index,[26 27] Timed Up and Go Test,[28] modified Patient-Specific Functional Scale,[29] 7-day objective physical activity levels using wrist-worn accelerometry (GENEActiv, Activinsights, Kimbolton, Cambridge UK) and a physical activity diary.

*Secondary outcomes*: Stroke Self-Efficacy Questionnaire,[30] Fatigue Assessment Scale,[31 32] Exercise Beliefs and Exercise Self-Efficacy questionnaires,[33] SF12,[34] EQ-5D-5L,[35] Stroke Quality of Life (QoL) questionnaires,[36] Carer Burden Index[37] and Health and Social Service use through a Service Receipt Inventory.[38]

Physical outcome baseline assessments (completed by research team) and follow-up assessments (at 6 and 9 months, completed by blinded assessor) were conducted in the participant's home. Researchers visited participants to fit the accelerometer, drop-off questionnaires and diary 1 week prior to blind assessor visits. Assessors administered primary outcome physical measures and collected accelerometers, questionnaires and diaries.

## Sample size

We required 48 participants (24 per group) as (1) 30 complete data sets are recommended for pilot studies to estimate outcome variance[39] and (2) we wanted to

investigate variations in context by running the intervention three times (*ie*, 3×8 patients). This number also allowed estimation of a predicted attrition rate of 20% with a precision of ±5% with 95% certainty.

## Randomisation and blinding

The random sequence was computer generated with minimisation for time since stroke (≤ 3 months vs > 3 months) and level of functional disability (modified Rankin Scale (mRS)[40] score ≤ 2 vs > 2). Allocation concealment was ensured by using a password protected validated web-based remote randomisation service supported by the Peninsula Clinical Trials Unit. The Trial Manager requested randomisation only after a cohort of participants had been consented.

Participants, trainers providing the intervention and researchers conducting the process and economic evaluations could not be blinded to allocation. However, outcomes were assessed by independent researchers (not based at research centre) who were blinded to group allocation. Participants were reminded not to reveal their allocation to assessors but any unblinding was recorded; after assessments assessors were asked to guess participant allocation.

## Data analysis

Analysis was primarily descriptive with participant flow summarised and estimates of screening, recruitment and attrition reported. Means and SD for all outcomes are reported at baseline, 6-month and 9-month follow-up for each group.

Intervention fidelity was assessed using mixed methods: qualitative video analysis comparing the trainer manual standard versus observed technique (two researchers independently assessed videos) combined with interview data and summary scores from trainer completed session checklists. Qualitative data were manually analysed descriptively and with content analysis for trial processes; additional thematic analysis was used for interview data. One person (MN) led the qualitative analysis but this was then discussed (MN and SGD), checked (RC) and agreed (MN, SGD, LP and RC).

We used a micro-costing approach to calculate costs associated with ReTrain: staff time (trainers, administrator and facilitators), venue hire, training equipment (annualised over time), course materials, consumables and travel costs (participants, trainers and facilitators). The costs of the intervention were estimated as a cost per programme and a cost per participant. The estimated costs of the intervention per participant were based on the number of participants enrolled on the programme. The base case scenario assumed the average number of participants per programme across all cohorts. Sensitivity analyses were conducted using the minimum and maximum number of participants enrolled for the programme and the quantity of programme materials that were wasted. We analysed the

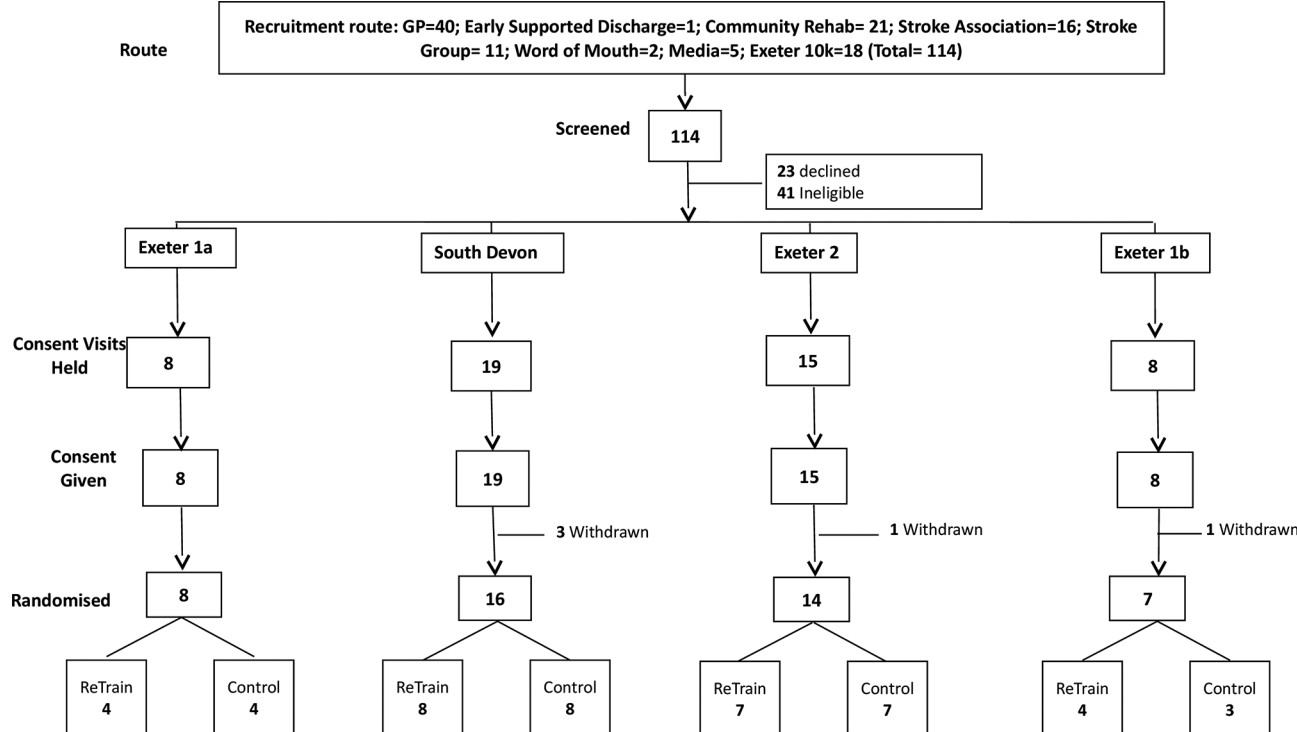

**Figure 1** Recruitment and randomisation by cohort.

relative benefits of calculating health-related QoL using SF-6D (developed from the SF-12) over the Quality-Adjusted Life Year (QALY) calculated (using EQ-5D 5L) from the baseline measures.

Sample size estimates for a definitive trial were calculated for candidate primary outcomes using the SD observed in this pilot population and published minimal clinical important difference (MCID) at 90% power and 5% alpha, and assuming 20% attrition. Where no published MCID could be sourced, we assumed a small to moderate effect size of 0.4 of a SD.[41] The trial statistician undertook calculations using the 'samspi' command in STATA V.14.2

## RESULTS

Recruitment took place from June 2015 to January 2016. The intervention ran in four cohorts, participant flows are shown for each (figure 1) and for the trial overall (figure 2). Initial recruitment was slow so to prevent late running of the trial we split the first cohort. Six-month follow-up outcome assessments took place in January to July 2016 and 9-month follow-up in April to October 2016.

### Objective 1: *Assess the feasibility and acceptability of recruitment, randomisation, allocation concealment and processes for outcome assessment and blinding*

We screened 115 individuals to recruit 50 participants (figure 1) in 8 months (2 months ahead of schedule). Of these, 45 (90%) were randomised (figures 1 and 2). Five individuals withdrew prior to randomisation due to ill health or the time lag between agreeing to take part and a cohort being ready to randomise. Table 1

shows baseline characteristics of those randomised, indicating a balance of characteristics across trial arms.

Blinding of outcome assessors was considered successful as only 2/41 (5%) participants revealed their allocations after completion of outcome measures, both were intervention participants. Different assessors were used for subsequent assessments therefore risk of bias was minimised.

### Objective 2: *Acquire retention rates and outcome variance*

Forty out of 45 (88%, 95% CI: 76% to 96%) completed 6-month and 9-month follow-ups. Despite fewer people being randomised than expected, high retention preserved the number of datasets needed to perform our sample size estimates (table 2).

### Objective 3: *Check ReTrain's acceptability and feasibility for participants and refine the trainer manual*

Eleven themes from 20 qualitative interviews summarise participants' views, box 1 provides illustrative quotes.
1. Study information: Participants considered information received as adequate. Five noted that information was limited, but most were unconcerned. Two added that too much information may have been detrimental to recruitment. Four others were satisfied with the information they received.
2. Outcome measure burden: Participants found the assessment process acceptable. Fifteen indicated no burden. Three participants indicated that they needed help from their carers to complete questionnaires, particularly recalling and reporting health resource use, placing a time burden on their carer.

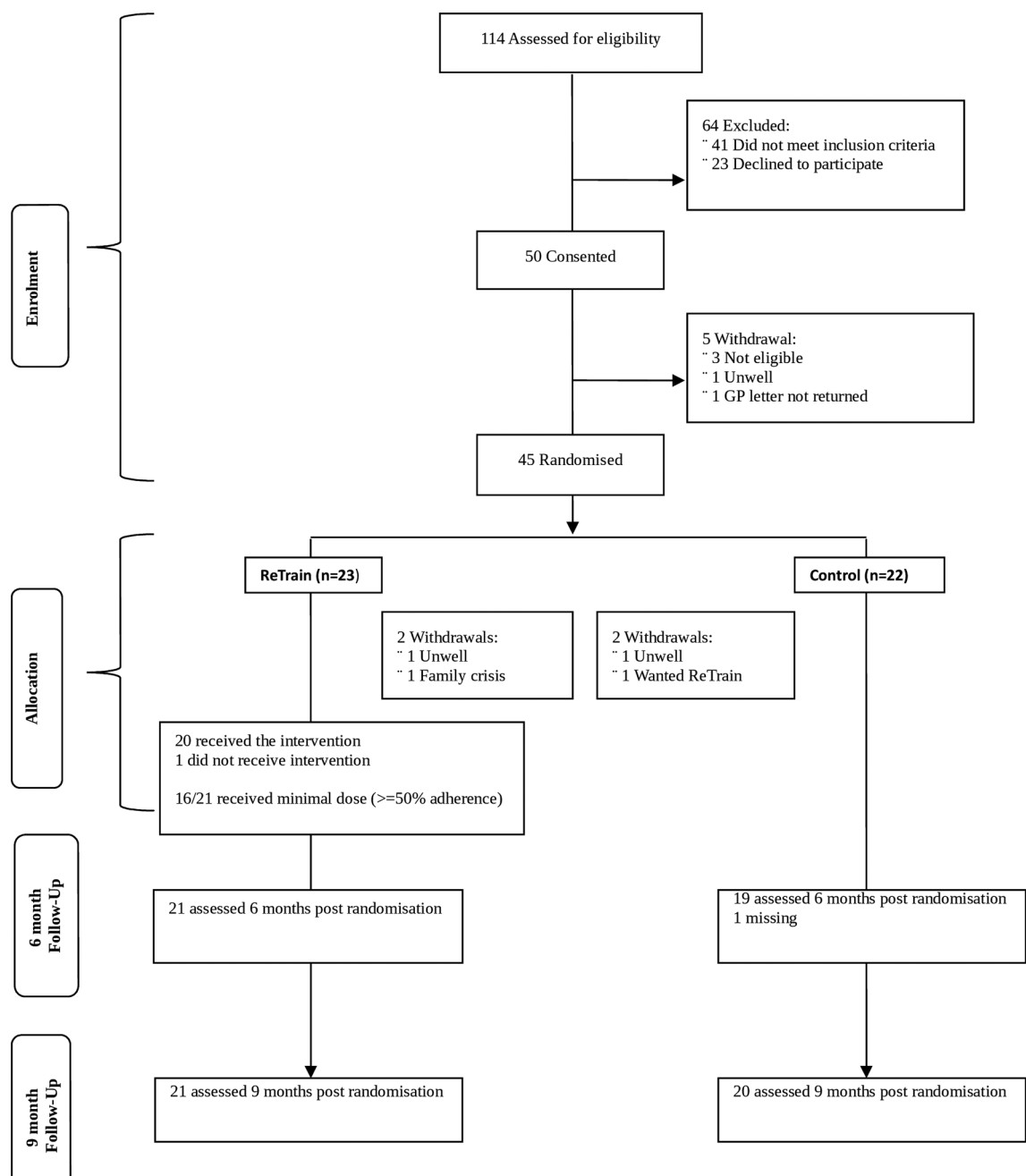

**Figure 2** Participant flow through the trial.

3. Venue: Half of the ReTrain participants were very positive about the training venues. Important features were space, provision of fluids (water and tea) and easy availability of parking. For some the travelling distance was a concern; two noted their venue (a gym) was very noisy, insufficiently heated and the session time was too early. Some noted the small amount of equipment as an advantage (it aided transfer of exercises to their home), whereas others felt that the equipment was not sufficiently specialist.

4. Adherence to ReTrain (see also Objective 5): All 10 ReTrain interviewees reported training in the 5 core (ARNI) techniques. Homework was discussed by all but lacked specificity, only two had clear homework

examples that were effectively incorporated into their training. Although goal setting was a core element, only four specifically identified how their goals were linked into their overall programme. Three participants reported not attending drop-in sessions due to lack of information. Of three who attended, two suggested the drop-ins repeated previous sessions.

5. Group dynamics: Group working was positively regarded and seen as integral to programme effectiveness. There were exceptions, one participant did not find 'performing' in public a positive experience. Likewise some suggested that groups reduced training intensity relative to one-to-one training.

| Table 1 Baseline participant demographics | | |
|---|---|---|
| | ReTrain (n=23) | Control (n=22) |
| Gender, n | | |
| Male (%) | 16 (70) | 14 (67) |
| Age, years, mean (SD) | 70 (12) | 71 (10) |
| Age category, years (n=45), n (%) | | |
| <45 | 1 (4) | 0 |
| 46–50 | 0 (0) | 1 (5) |
| 51–60 | 3 (13) | 2 (9) |
| 61–70 | 10 (43) | 6 (27) |
| 71–80 | 5 (22) | 8 (36) |
| 81–90 | 2 (9) | 5 (23) |
| 90+ | 2 (9) | 0 (0) |
| Time since stroke, months, n (%) | | |
| <12 | 3 (13) | 3 (14) |
| 12–24 | 4 (17) | 4 (18) |
| 25–48 | 5 (22) | 5 (23) |
| 49–72 | 2 (9) | 5 (23) |
| 73–96 | 4 (17) | 2 (9) |
| 97+ | 5 (22) | 3 (14) |
| Time since stroke minimisation categories, months, n (%) | | |
| ≤3 | 1 (4) | 0 (0) |
| >3 | 22 (96) | 22 (100) |
| Type of stroke, n (%) | | |
| Haemorrhagic | 3 (13) | 1 (5) |
| Ischaemic | 15 (65) | 15 (68) |
| Both | 0 (0) | 1 (5) |
| Missing | 5 (22) | 5 (23) |
| Stroke rehabilitation, weeks | | |
| n | 21 | 21 |
| Average no of weeks (SD) | 8 (9) | 14 (19) |
| Median no of weeks | 6 | 12 |
| Range | 0–32 | 0–88 |
| Unknown length rehab, n | 2 | 1 |
| Functional disability (sMRS), n (%) | | |
| 0 | 1 (4) | 0 (0) |
| 1 | 2 (9) | 1 (5) |
| 2 | 4 (17) | 9 (41) |
| 3 | 16 (70) | 12 (55) |
| sMRS minimisation categories, n (%) | | |
| ≤2 | 7 (30) | 10 (45) |
| >2 | 16 (70) | 12 (55) |
| Comorbidities,* n (%) | | |
| Hypertension | 18 (78) | 18 (82) |
| Type 2 diabetes mellitus | 4 (17) | 4 (18) |
| Depression | 8 (35) | 5 (23) |

Continued

| Table 1 Continued | | |
|---|---|---|
| | ReTrain (n=23) | Control (n=22) |
| Chronic kidney disease | 2 (9) | 1 (4) |
| Asthma/COPD | 4 (17) | 3 (14) |
| Other | 5 (22) | 3 (14) |
| Medications,* n (%) | | |
| Diuretics | 3 (13) | 1 (5) |
| Anticoagulants | 8 (35) | 10 (45) |
| Antiplatelet | 15 (65) | 12 (55) |
| Antihypertensives | | |
| Calcium channel blockers | 6 (26) | 14 (64) |
| ACE inhibitors | 13 (57) | 8 (36) |
| Other | 9 (39) | 7 (32) |
| Statins (%) | 18 (78) | 19 (86) |
| Antidepressants (%) | 8 (35) | 5 (23) |
| Diabetes medication (%) | 4 (17) | 4 (18) |
| Chronic pain medication (%) | 12 (52) | 8 (36) |
| Other (%) | 5 (22) | 3 (14) |
| Employment status, n (%) | | |
| Employed (and working) | 2 (9) | 1 (5) |
| Retired | 18 (78) | 15 (68) |
| Semiretired | 1 (4) | 0 (0) |
| Unemployed | 2 (9) | 5 (27) |
| Prestroke exercise history, n | | |
| Exerciser (%) | 10 (43) | 8 (36) |
| MMSE | | |
| n | 22,† | 22 |
| Mean (SD) | 27.5 (2.54) | 27.9 (3.01) |
| Median | 28 | 29 |
| Range | 19–30‡ | 19–30‡ |

*Participants may have more than one comorbidity/medication.
†One participant with severe aphasia had difficulties completing the MMSE. The participant could understand and follow instructions and was considered cognitively able to participate in the trial.
‡Higher scores indicate better cognitive function. Participants range from no to moderate degree of cognitive impairment.
ACE, angiotensin-converting enzyme; COPD, Chronic obstructive pulmonary disease; MMSE, Mini–mental state examination; ReTrain, Rehabilitation Training; sMRS, Simplified Modified Rankin Scale score.

6. Comorbidities: Participants identified several comorbidities, such as knee replacements, cancer, angina, diabetes, amputation and depression. These had potential to impact both the training and research participation but for most any concerns were accommodated by trainers. However, in one case, some uncomfortable discussions occurred before an appropriate

**Table 2** Sample estimates for potential candidate primary outcomes from ReTrain pilot RCT

| Primary outcome measure | Sample size estimates* | MCID | Observed SD range | Effect size (MCID/SD) |
|---|---|---|---|---|
| Rivermead Mobility Index | 36–44 | 3.0† | 2.33–2.66 | 1.13–1.29 |
| Timed Up and Go | 1438–2673 | 1.2 – 3.4‡ | 15.69–21.39 | 0.06–0.22 |
| Modified Patient-Specific Functional Scale | 16–200 | 1.0 – 3.0‡ | 1.58–1.94 | 0.52–1.7 |
| PA (accelerometer) | 350–1458 | Not available | Not applicable | 0.2–0.45§ |

*Figures represent overall (two groups) sample size estimates required for a definitive trial. Sample sizes estimated for 1:1 allocation at 90% power and 5% alpha and assuming 20% attrition. Calculations are conservative showing range from best case scenario (largest MCID and smallest SD) to worst case scenario (smallest MCID and largest SD) of SDs observed in this trial and published MCIDs where available.
†MCID available from stroke research for the Rivermead Mobility Index (http://www.strokengine.ca/psycho/rmi_psycho/).
‡MCIDs identified from other disease groups used as proxies as no published stroke MCIDs.[47 48]
§There are no MCID data available for PA (accelerometry) in stroke (or any other cardio vascular disease), we therefore applied sample size calculations undertaken for a relevant ongoing HTA NIHR trial, which estimated n=562 (effect size 0.3) or n=413 (effect size 0.35) (http://www.isrctn.com/ISRCTN15644451).
HTA NIHR, Health Technology Assessment, National Institute of Health Research; MCID, minimal clinical important difference; PA, physical activity; RCT, randomised controlled trial; ReTrain, rehabilitation training.

balance of perceived capability and training challenges was reached. Three participants with visual deficits, dyslexia and dysgraphia mentioned difficulties in completing the research documents.

7. Carer health: Two ReTrain participants commented on how commitment to the programme impacted their partner's health: one stopped attending sessions because the time away resulted in excessive strain on his wife; another expressed similar concern but did not stop attending.

8. Trainer manual: we refined the trainer manual throughout the study. Issues raised during interviews guided revisions including greater emphasis and clarification about use of goal setting, drop-ins, homework diaries and managing participants with comorbidities.

**Objective 4: *Assess outcome completion and burden***
We collected baseline (n=41), 6-month (n=40) and 9-month (n=41) follow-up data on the majority of participants (figure 2). Accelerometry wear time (24 hours for 7 days) was high, most having 6 or more valid days (≥16 hours per day, including ≥1 weekend day). Only two participants at baseline, one at 6 months and three at 9 months did not achieve 4 valid days of wear time. There was very little missing data. For three primary outcome measures, there was only one participant with missing data at any given assessment time-point. For secondary outcomes, there was either no missing data or only one to two participants with missing data at each time-point, apart from the exercise diary (between two and four participants with missing data at each time-point) and the service receipt inventory (between three and seven participants with missing data). There were eight participants without accelerometer data at 9-month assessment owing to hardware (device) and software (data extraction method) malfunctions.

**Objective 5: *Perform process evaluation with an assessment of intervention fidelity***
We implemented a comprehensive video recording schedule (over 200 recordings) to capture participant and trainer adherence to key ARNI techniques. Both trainers and participants demonstrated high adherence. Modifications to techniques (to accommodate participant comorbidities) were captured and informed trainer manual development.

We combined metrics from attendance registers and homework records to generate a 'dose'/adherence score, categorising individuals into low (<50%), medium (50%–75%) and high (>75%) adherence categories. Of 23 ReTrain participants, 2 did not receive the intervention (1 returned to work; 1 withdrew from the study), 5 had low adherence, 5 medium adherence and 11 high adherence. These latter 16 (70%) were considered to have received sufficient 'dose' of ReTrain.

Trainers varied in their completion of session checklists: pre-exercise and end-of-session components were less consistently reported compared with ARNI techniques but overall there was good adherence to programme delivery.

**Objective 6: *Calculate the cost of intervention delivery and feasibility of collecting health and social service resource use***
ReTrain costs were generated for each cohort, accounting for different programme sizes (four or eight participants) and venues. Costs per participant ranged from £615 to £972. The total per participant cost for ReTrain (assuming 24 participants) was £777. We conducted medical notes review on 35/41 participants and compared this 'gold standard' with self-reported health resource use. Participants reported using fewer resources compared with case notes review. Data from medical notes informed the cost–utility and effectiveness frameworks for use in a definitive trial.

## Box 1 Participant quotes from qualitative interviews

**Acceptability**

"It is ten weeks, you do it twice a week. Personally for the first say three or four weeks, I'd think well this is getting me nowhere, but then you think that you notice things, things are improving and at the end of ten weeks you want to go for twenty weeks" (4; 119–125).

"I'd tell them [another stroke survivor] to go ahead and do it and to take it step by step and not to worry about it. Because you are treated with great respect, it was wonderful and they were. I'll never be able to speak highly enough of them." (25; 388–390).

**Intervention approach**

"It opened my eyes to what can be done you know. How can I put it? It wasn't as if I believed that I couldn't do something it was being pointed in the right direction…heh I can do it……Great you've done it, you did it and you do it again. Yeah it was great" (4; 358–361).

"It wasn't easy at first, but I used to manage it" (5; 246).

"It was the way they addressed how you do your exercises. What it is doing to you and all the rest of it. Now to me that was absolutely important, because it made sense of why you are doing all this pumping up and down, and if you can't do that, do this" (22; 252–255).

"It was you felt as if you were a human being with them. You know and you were treated with respect…and although you couldn't do things and you felt a bit of an idiot, they never let you feel like that" (25; 567–572).

"It's a bit like playing scales…it's not creative but as I gradually realise it, it could potentially be creative…doing something that I had been doing without thinking before and now couldn't. …Now and again I walk without my stick without realising it, that's creative I think" (6; 354–392).

**Impact of programme: psychologically**

"I suppose it is attitude of mind as much as anything. I mean I felt I'd gone through that stage of training and that I was going to get better. It built my spirit up…I felt as if it, well it was worth the three months you know and at the end of the day I hope I'm going to get back to something like normal" (16; 358–365).

"It really helped me mentally, you know I thought right I can do this because before I was going into my shell, thinking I can't do this and I can't do that. Oh I am not going out. Then I went on that [ReTrain] and it gave me an element of confidence" (43; 562–565).

**Impact of programme: physically**

"You started to notice they are actually starting to fall into place. I don't remember doing that last time. But I am doing it now great get on with it I am doing it faster now" (4; 189–190).

"I know if I went down which I did one day in the hall in the early stages of coming back home and I did manage to get up and walk upstairs…but I wouldn't have been able to do that had I not had that [training]" (16; 475–477).

**Homework adherence**

"Trainers were always on about doings exercises at home…I could never pin him down to how long that should be for though" (6; 624).

**Programme technique adherence**

"I think that was the big thing you saw the benefits after the second, well the first or second session we had.' 'Oh we can do."

### Descriptive analysis of participant outcomes

Table 3A,B report number, mean scores and SDs across, respectively, candidate primary and secondary outcome measures at each time point of the pilot trial. The trial was not powered to detect differences in outcome between trial arms or over time and so we do not interpret the patterns of means. However, the results clearly demonstrate that we were able to collect the necessary data and retained acceptable completion rates on all measures across all time points of the study. Attrition was lower than the 20% expected. For each outcome measure (except carer burden as not everyone had a carer), we achieved in excess of the 30 cases (ie, 15 completed measurements per arm) recommended for pilot studies to estimate outcome variance.

### Safety

During assessment periods, there was one serious but unrelated event in the intervention group (none in the control group) and slightly fewer overall adverse events in the intervention group (table 4A).

For ReTrain only (table 4B), there were six serious adverse events during the intervention period: four were unrelated, one possibly related (fainted) and one probably related (TIA) to the intervention. Of the 22 adverse events reported, 3 of them occurred at the venue (one x fall; one x trip; one x ankle strain).

### DISCUSSION

The ReTrain pilot trial met all its prestated feasibility objectives: the intervention, trial design and research

**Table 3A** Number, means and (SD) as a function of trial arm and measurement time point for candidate primary outcome measures in the ReTrain pilot trial

| Measures, n, mean (SD) | Data collection time point | | | | | |
| --- | --- | --- | --- | --- | --- | --- |
| | Baseline | | 6months* | | 9months* | |
| | ReTrain (n=23) | Control (n=22) | ReTrain (n=21) | Control (n=22) | ReTrain (n=21) | Control (n=20) |
| Rivermead Mobility Index | 23, 11.41 (3.05) | 22, 11.68 (2.23) | 21, 12.14 (2.73) | 19, 12.47 (1.87) | 21, 12.24 (3.27) | 20, 12.65 (1.81) |
| Modified Patient-Specific Functional Scale | 22, 2.95 (1.85) | 22, 2.55 (1.23) | 21, 3.47 (2.12) | 19, 3.56 (1.69) | 21, 3.25 (2.03) | 20, 3.74 (1.86) |
| Timed Up and Go (s)† | 23, 27.57 (27.57) | 21, 21.24 (11.18) | 21, 20.76 (19.64) | 19, 16.37 (9.69) | 21, 20.76 (19.25) | 20, 15.95 (12.00) |
| PA: diary‡ | 21, 6.67 (19.20) | 20, 10.69 (17.39) | 21, 17.39 (24.28) | 19, 25.60 (34.98) | 19, 13.92 (22.25) | 20, 35.11 (49.70) |
| PA (accelerometer): total PA min‡§ | 21, 145.10 (118.27) | 20, 165.56 (139.09) | 19, 134.88 (129.77) | 18, 178.15 (155.07) | 16, 152.08 (118.52) | 17, 197.42 (144.31) |
| PA (accelerometer): light PA min‡§ | 21, 92.78 (93.15) | 20, 110.18 (115.65) | 19, 95.67 (105.50) | 18, 121.80 (122.46) | 16, 99.33 (99.54) | 17, 134.54 (126.36) |
| PA (accelerometer): MVPA PA min‡§ | 21, 52.32 (68.94) | 20, 55.38 (39.66) | 19, 39.21 (39.33) | 18, 56.35 (51.26) | 16, 52.75 (60.03) | 17, 62.88 (41.73) |
| PA (accelerometer): moderate PA min‡§ | 21, 50.53 (66.77) | 20, 53.38 (37.04) | 19 37.73 (37.40) | 18, 53.07 (43.99) | 16, 51.11 (58.54) | 17, 60.93 (40.84) |
| PA (accelerometer): vigorous PA min‡§ | 21, 1.79 (3.85) | 20, 2.00 (3.96) | 19, 1.48 (2.39) | 18, 3.28 (8.14) | 16, 1.64 (2.38) | 17, 1.94 (2.33) |

*Postrandomisation.
†Precision to 10 ms.
‡Average minutes of physical activity per day.
§Measurement recorded 100 times a second (accelerometer set to a sampling frequency of 100 Hz).
MVPA, Moderate to Vigorous Physical Activity; PA, physical activity; ReTrain, rehabilitation training.

**Table 3B** Number, means and (SD) as a function of trial arm and measurement time point for candidate secondary outcome measures in the ReTrain pilot trial

| Measures, n, mean (SD) | Data collection time point | | | | | |
| --- | --- | --- | --- | --- | --- | --- |
| | Baseline | | 6months* | | 9months* | |
| | ReTrain (n=23) | Control (n=22) | ReTrain (n=21) | Control (n=20) | ReTrain (n=21) | Control (n=20) |
| Fatigue Assessment Scale | 23, 27.17 (7.14) | 22, 25.49 (7.44) | 21, 24.05 (6.26) | 19, 24.05 (8.86) | 20, 27.45 (5.95) | 20, 25.83 (9.14) |
| Stroke Self-Efficacy Questionnaire | 22, 72.41 (22.00) | 22, 73.46 (17.87) | – | – | 20, 73.73 (19.63) | 20, 74.40 (16.94) |
| Outcome expectations for exercise scale (Exercise Beliefs) | 23, 3.66 (0.70) | 22, 3.78 (0.52) | – | – | 19, 4.03 (0.59) | 19, 3.73 (0.52) |
| Short Self-Efficacy for Exercise Scale (Exercise Self-Efficacy) | 23, 3.26 (0.92) | 22, 3.32 (0.89) | – | – | 19, 3.32 (0.89) | 18, 3.22 (1.06) |
| Stroke QoL Scale (total) | 22, 3.31 (0.68) | 22, 3.45 (0.69) | – | – | 20, 3.38 (0.70) | 20, 3.63 (0.82) |
| EQ-5D-5L | 22, 0.51 (0.25) | 20, 0.55 (0.24) | – | – | 19, 0.52 (0.24) | 20, 0.62 (0.25) |
| SF-12: physical component | 21, 33.12 (7.22) | 20, 31.83 (6.69) | – | – | 19, 33.74 (6.44) | 19, 33.25 (6.91) |
| SF-12: mental component | 21, 50.10 (7.11) | 20, 50.68 (7.98) | – | – | 19, 50.47 (6.51) | 19, 48.05 (8.45) |
| Modified Caregiver Strain Index (carer burden) | 8, 11.39 (8.03) | 10, 7.40 (7.83) | – | – | 9, 9.89 (7.22) | 6, 9.50 (8.92) |

-, indicates measurement not taken at this time point.
*Postrandomisation.
ED-5D-5L, EuroQol 5 Dimension, 5 Level, measure of health-related Quality of life; QoL, quality of life; ReTrain, rehabilitation training; SF-12, 12 item Short-Form Health Survey; SF-6D, Short-Form, Six Dimension Health Index.

| Table 4A | Adverse Events and Serious Adverse Events reported during 6-month and 9-month outcome assessment periods for both ReTrain and control groups | | | | | | |
|---|---|---|---|---|---|---|---|
| | | | **Attribution** | | | | **No of people reporting event** |
| | Event type | Total events | Related | Probably related | Possible related | Unrelated | |
| ReTrain (n=21) | AE | 125* | 6 | 5 | 73 | 41 | 19 |
| | SAE | 1† | 0 | 0 | 0 | 1 | 1 |
| Control (n=20) | AE | 150‡ | 0 | 0 | 0 | 150 | 19 |
| | SAE | 0 | 0 | 0 | 0 | 0 | 0 |

*Muscle soreness (n=26), fatigue (n=58), falls (n=12), trips (n=10) and other (n=19; including but not limited to: low mood, itchiness, colds, issues with eyesight and cystitis).
†Ambulance conveyance to A&E due to reaction to antibiotics being taken for chest infection.
‡Muscle soreness (n=39), fatigue (n=50), falls (n=19), trips (n=12) and other (n=30; including but not limited to: low mood, depression, dizzy spells, sore toes, poor memory, colds, poor sleep, loss of sense of smell and issues with eyesight).
AE, adverse events; ReTrain, rehabilitation training; SAE, serious adverse events.

processes were acceptable to participants as well as feasible and safe to deliver; we demonstrated feasibility of recruitment (recruiting above our target of 48) and retention (<20% attrition). At the point of randomisation, we were slightly under target (45/48). However, due to high retention, we preserved the number of datasets required (30) to calculate sample size estimates. Furthermore, participants were not unduly burdened by study requirements and there were high completion rates for most outcome measures. We also successfully rehearsed procedures for process and health economic evaluations as well as trial governance processes (trial management and independent trial steering meetings) and maintained our strong Patient and Public Involvement. Participant interviews, outcome measurement results and fidelity assessments highlighted refinements that we have already, or can, put in place for a future definitive RCT of ReTrain. For example, we have some new insights into how to enhance delivery by trainers and engagement by participants (eg, by placing more focus on individually tailored goal setting; stressing goal and homework reviews; better explanation and promotion of the drop-in sessions). These are all relatively small amendments that are likely to enhance the impact of the training programme. Our trial compares favourably with another feasibility RCT assessing the delivery of the Bridges stroke self-management programme,[42] which had relatively low recruitment, questions regarding programme delivery in addition to usual rehabilitation and recommendations for further assessment of intervention fidelity. Some of their findings were similar to ReTrain: participants were broadly positive about their programme; health professionals found it acceptable to use and researchers noted the lack of outcome measure sensitivity for detecting change.[42]

### Limitations and lessons for planning design of a future trial

When planning this study we selected our candidate primary outcome measures on the basis that they were likely to measure improvements that could be attributed to our intervention; our pilot work was therefore to determine acceptability and feasibility (including their psychometric utility) of these measures. However, we were not able to identify a clear candidate primary outcome for a definitive RCT from this pilot work. It is possible that an 'activities of daily living' measure (as typically used in rehabilitation studies) may be more useful in a future definitive trial. Identifying robust outcome measures in rehabilitation trials is a common problem,[43] compounded by variability in stroke-related disability and participants' comorbidities. This pilot trial was not designed (statistically powered) to test for differences between treatment arms, so no inferential analyses were performed. Any perceived trend (or absence of a trend) should not be interpreted as an indication of an effect (or its absence) and outcomes should not be selected based on any assumed trend. Acceptability outcomes coupled with a pragmatic and efficient (cost-effective) trial design better inform choice

| Table 4B | Adverse Events and Serious Adverse Events reported during ReTrain programme | | | | | | |
|---|---|---|---|---|---|---|---|
| | | | **Attribution** | | | | **No of people reporting event** |
| | Event type | Total events | Related | Probably related | Possible related | Unrelated | |
| ReTrain (n=21) | AE | 2* | 7 | 0 | 12 | 3 | 11 |
| | SAE | 6† | 0 | 1 | 1 | 4 | 5 |

*Muscle soreness (n=0), fatigue (n=2), falls (n=10), trips (n=1) and other (n=9; including but not limited to: fainting; twisted or swollen ankle and suspected TIA (non-confirmed)).
†Urine retention (n=3), black-out/fainted (n=1), renal and heart failure (n=1) and TIA (n=1).
AE, adverse events; ReTrain, rehabilitation training, SAE, serious adverse events; TIA, Transient Ischaemic Attack.

of outcome. From our sample, the Timed Up and Go task would be unsuitable due to potentially large sample size requirements (~2000 participants) and the baseline high levels of mobility meant that the Rivermead Mobility Index demonstrated a ceiling effect, so could only be used if we altered inclusion criteria. Physical activity was measured robustly via accelerometry and may be the best candidate. We had some software and hardware malfunctions but important lessons have been learned to mitigate these problems in future. Capture of frequency and intensity of activity would allow comparison with stroke guidelines. Although there is a cost implication, accelerometry provides a more objective measurement of daily activity and may also be an adequate proxy of functional mobility; however, we will also investigate the benefits of using other PA measures such as questionnaires (instead of our diaries) or using multiple measures such as accelerometry and heart rate monitors while being aware of problems with compliance and participant burden.[44]

Further limitations relate to the lack of validation of our adherence measure and the local demographics: our sample did not have a wide age range or ethnic diversity. While we did demonstrate delivery in different locations in the South West, our plans for a larger definitive trial would include a wider demographic from more centres across the UK.

For a future trial, we plan to implement more readable, higher quality written (and pictorial) information and questionnaires although the amount of information provided was appropriate. We will mitigate recruitment loss prior to randomisation by establishing expression of interest and eligibility to take part but delaying taking consent until we are confident of sufficient numbers to create a cohort for randomisation; this has resource implications that will need to be built into future funding. We will run ReTrain in community centres or halls as these were more acceptable and much cheaper than gyms; we will provide a more detailed ReTrain induction to ensure trainers understand and communicate all components of the programme. For the QALY comparisons, recent policy changes mean the conversion from SF-12 to SF-6D has been phased out, and so less justification for using the SF-12 in a future study. Instead we will consider using the Stroke Impact Scale (SIS) as this is a valid health-related QoL measure. This may also be a better candidate self-report primary outcome measure for a definitive trial as it has shown sensitivity in long-term stroke survivors who have mild to moderate stroke.[45] The SIS assesses multiple facets of physical and emotional issues and so would align with perceived physical and psychological benefits participants attribute to ReTrain. Our sample size estimates for candidate objective primary outcomes (table 2) indicate that we will need a moderately sized trial (n=562, effect size 0.3 or n=413, effect size 0.35) for PA assessed by accelerometry or a smaller trial (n=96)

if we use the physical component domain of the SIS (based on 80% power, 5% alpha and assuming 20% attrition[46]). We have established appropriate process evaluation methods to capture multiple facets of intervention fidelity.

## Generalisability

This pilot study was not designed to demonstrate generalisability; however, our participant population represent the subset of community-dwelling stroke survivors who have some independent mobility but remain with stroke-related disability that affects their QoL. Our participants also represent the growing proportion of people who have more than one long-term condition. ReTrain techniques target the effects of stroke but can accommodate other conditions which trainers take into account when preparing the participant's individually tailored programme. Some of the key ReTrain (ARNI-based) techniques are designed for people with unilateral impairment, such as hemiparesis; however, one of our participant's main unilateral impairment was due to diabetes-related lower limb amputation, illustrating how ReTrain can accommodate people with multiple comorbidities.

## CONCLUSION

Our pilot trial has demonstrated that ReTrain is feasible, acceptable and safe. We met our recruitment and retention targets and demonstrated our ability to run our intervention in different locations. Participants were not unduly burdened by study requirements and most outcome measures had high levels of completion. We successfully tested procedures for process and health economic evaluations. Participant interviews, outcome measurement results and fidelity assessments highlighted some issues needing refinement prior to a future definitive RCT of ReTrain. Many of these have already been addressed and we intend to seek funding for a definitive trial.

**Author affiliations**
[1]University of Exeter Medical School, Exeter, UK
[2]Academic Unit of Elderly Care and Rehabilitation, University of Leeds, Bradford, UK
[3]Royal Devon and Exeter NHS Foundation Trust
[4]Torbay and Southern Devon NHS Foundation Trust, Torquay, UK
[5]Department of Clinical Sciences, Brunel University London, London, UK
[6]Department of Sport and Exercise Science, University of Portsmouth, Portsmouth, UK
[7]Sports and Health Science, University of Exeter, Exeter, UK
[8]Royal Devon and Exeter NHS Foundation Trust, Exeter, UK

**Acknowledgements** We thank our funders, the Stroke Association and the Peninsula Patient Involvement Group with the ReTrain Stroke Service User Group for their help. The National Institute for Health Research (NIHR) Collaboration for Leadership in Applied Health Research and Care South West Peninsula at the Royal Devon and Exeter NHS Foundation Trust also supported this work but views expressed are those of the author(s) and not necessarily those of the NHS, the NIHR or the Department of Health. We also thank our Trial Steering Committee: Ailie Turton (University of the West of England) Siobhan Creanor (Plymouth University), Debbie Neal (Bournemouth University), Justin Smallwood (Patient and Public

representative) and Gail Seymour (University of Exeter - Sponsor). Ethical review by NRES Committee South West – Cornwall & Plymouth (REC ref: 15/SW/04).

**Contributors** SGD: led the team and drafted this article. RC: prepared protocol, ethical submission and amendments, managed the project and contributed to analysis. LP: drafted protocol prior to funding application, conducted interviews and contributed to analysis. AF, MJ, RA, MN, SGD and LP: provided stroke rehabilitation expertise. RST: provided statistical and trial methodological expertise, led analysis. MN: provided qualitative expertise and analysed qualitative data. AIS: led accelerometry work, supported by RMP who provided accelerometry analysis. SGD and LP: provided process evaluation expertise. SGD: led the process evaluation and supervised LH. LH: led video analysis work. AS: provided health economic expertise and led economic work supported by PL. SS: provided patient and public involvement expertise. All authors commented on the manuscript.

**Funding** The Stroke Association TSA 2014-13.

**Competing interests** Dean reports grants from The Stroke Association, during the conduct of the study; other from Wiley Blackwell, outside the submitted work; Forster, Poltawski, Spencer, Taylor, James, Allison, Stevens, Pulsford and Norris report grants from the Stroke Association; Calitri, Landa, Hollands, Shepherd were employed by the Stroke Association grant.

**Patient consent** Obtained.

**Ethics approval** NRES Committee South West—Cornwall & Plymouth (REC ref: 15/SW/04).

**Provenance and peer review** Not commissioned; externally peer reviewed.

**Data sharing statement** Participants did not consent for datasets to be stored or accessed outside of the research team. Therefore, no datasets have been made publicly available.

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
