## [Reviewer comments · BMJ Open]

ARTICLE DETAILS

TITLE (PROVISIONAL)	Community-based Rehabilitation Training after stroke: Results of a pilot randomised controlled trial (ReTrain) investigating acceptability and feasibility
AUTHORS	Dean, Sarah; Poltawski, Leon; Forster, Anne; Taylor, Rod; Spencer, Anne; James, Martin; Allison, Rhoda; Stevens, Shirley; Norris, Meriel; Shepherd, Anthony; Landa, P; Pulsford, Richard; Hollands, Laura; Calitri, Raff

VERSION 1 – REVIEW

REVIEWER	Prof Marion Walker University of Nottingham, England
REVIEW RETURNED	17-Jul-2017

GENERAL COMMENTS	Thank you for asking me to review this paper which I enjoyed reading very much. This was a very well thought through study resulting in a well written paper with comprehensive reporting. I have a few minor comments to make: Patients recruited from one month post stroke – in reality few patients were recruited very early after stroke but some consideration for a future trial should be given if patients are still receiving ongoing rehabilitation such as ESD or community rehabilitation. Adverse events were recorded and reported – it would be helpful to know what these consisted of - particularly adverse events at the venue. Very many outcome measures applied which I imagine placed significant burden to patient and carer. For me one of the biggest limitations of the study was the lack of a robust ADL measure. It would have been extremely helpful if a measure of extended ADL had been recorded, eg the Nottingham Extended ADL. The study used the Patients Specific Functional Scale. This is a self-reported scale and not a commonly used scale in stroke rehabilitation research. It would have been hugely helpful if the authors had used an ADL measure that would allow comparison with other stroke rehab trials – for future meta-analyses. I look forward to seeing this study progress to a large scale multi-centre study.
--

REVIEWER	Elissa Burton Curtin University Australia
REVIEW RETURNED	19-Jul-2017

GENERAL COMMENTS	The aims of the article were to assess the acceptability and feasibility of trial processes and the ReTrain intervention including an assessment of intervention fidelity. This article was extremely well written and of high quality. I recommend only a few minor revisions which the authors may like to consider. Title: The title suggests it is the results of a pilot RCT yet the paper really is more than that and looks at feasibility and acceptability. Perhaps consider altering this to make it clearer to readers. Abstract: Line 16-20: eligibility should there be some note about having had a stroke? Line 19 spell out NHS first time Introduction: Page 4, Line 7 UK spell out fully first time Methods Page 6, lines 36-39 need semi-colons rather than commas to be consistent with above primary outcomes Page 7 randomisation and blinding. Who actually created the randomisation? (It does say in the protocol, not sure if you want to add to this paper as well). Page 7, was any computer package used to analyse the qual data? (eg NVivo). How many people analysed the qual data? How was content analysis conducted as well as the thematic analysis? Needs a little more detail. Economic evaluation would also benefit with more detail on how you came up with the results. Ethics approval did not seem clear in the paper but it was suggested in the acknowledgements, it would be better to be clearer on this. Results Page 8, Table 1 for gender remove the % for 70% or add it into 67. Include standard deviation after the mean age in years. Where there are mean values throughout the table I would include SD as you did in the MMSE. Page 10, Objective 3, line 40 the heading says acceptability and feasibility for participants and complete the Trainer manual, should the and be to complete? The and doesn't sound quite right Page 10, line 55. How did the questionnaires place burden on the carer? Page 14 and 15, Table 4a and Table 4b: It would be good to have a few sentences actually describing these two tables. Page 16, the type of adverse or serious adverse events should be listed Discussion Page 17, lines 15-17 would be good to actually state what some of the changes, refinements were rather than having to look through and try and work out which were changed etc Page 18, line 14-15, what changes were made to the manual (state if they are large and how they affect the programme) References: Number 15: add r to eport
---

REVIEWER	Avril Mansfield University Health Network, Canada
REVIEW RETURNED	25-Jul-2017

GENERAL COMMENTS	Overview This paper describes a pilot study of an intervention including community-based rehabilitation among individuals with stroke. The investigators assessed feasibility of a larger trial by describing rates of recruiting and attrition, participants' acceptability of the intervention, estimating sample size required for a larger trial, and estimating costs associated with the intervention. Major revisions 1. The primary outcomes identified in the trial registration differ from those included in this paper, with the outcomes in the paper being focused on addressing feasibility objectives. While I agree that, as this is a pilot study, the outcomes should address feasibility issues, the authors must provide some justification for altering the plan outlined in the trial registration. 2. There should be some hypotheses around the objectives identified in the last paragraph of Page 4. The authors conclude, in the 1st sentence of the Discussion, that the trial met its feasibility objectives. However, it is not clear how they arrived at this conclusion. For example, for objective 1, what recruitment numbers, randomisation rates etc would be acceptable to establish feasibility of a larger trial? 3. More information about the experimental and control interventions are required. Consider the FITT principle (frequency, intensity, type, and time) when describing the exercise portion of the intervention; e.g., what specific exercises were included, how many repetitions/sets, how was intensity set (% of one-repetition maximum?), how was exercise dose progressed through the program? What was discussed in the home visits? For the control group, what was 'usual treatment'? These participants were discharged from the NHS rehabilitation services, so it seems that there was no treatment available to these participants; is this the case? Did the investigators document any treatment that control participants received outside of the trial (e.g., private rehabilitation services)? 4. There is no statement regarding research ethics approval or participant consent in the paper. Minor revisions 5. Please provide justification for using minimization to allocate participants rather than simple or stratified randomization. 6. Table 2: please include the MCIDs (or biologically/clinically meaningful effect sizes) and the standard deviation/variance used to calculate the sample sizes. Please clarify if this is the number of participants required per group, or the overall sample size for both groups combined. Why are ranges of sample sizes presented for some variables? 7. Tables 4a and 4b: the captions state that means and standard deviations are presented, but there are three numbers in each cell; what is the third number?
--

Also, please consider the precision with which numbers are presented; for example, was the Timed Up and Go measured with 10 ms precision?

8. Table 4b: include the name of the specific scales used to measure stroke & exercise self-efficacy, exercise beliefs, and quality of life.

9. Page 16, Line 7: replace “less” with “fewer”.

10. ‘Safety’ section: please provide further details regarding the nature of the adverse events (e.g., events possibly/probably/definitely related to the intervention might include muscle soreness or joint pain during/following exercise). Please clarify why some AEs are classified in Tables 5a and 5b as being ‘related’ when, in the text, it is stated that no AEs were definitely related to the intervention.

11. I do not understand the argument presented in the second paragraph of the Discussion regarding selection of a primary outcome measure for the larger trial. The investigators should select an outcome that measures what they would like to improve with the intervention (i.e., the target of the intervention). From the objectives presented on Page 5, the intervention aims to: enhance “function”, develop self-management skills, and instill commitment to exercise. Therefore, the larger trial should assess if the intervention meets these objectives by including outcomes that measure “function”, self-management skills, and adherence to exercise in daily life. Selecting outcome measures based *solely* on characteristics of outcomes measured during the pilot study (e.g., ceiling effects, high variability) does not make sense.

12. Regarding measurement of physical activity, several investigators have suggested combining data from heart rate monitors, accelerometers, and physical activity questionnaires among individuals with stroke (e.g., see Resnick et al., *J Phys Act Health*, 2008; Zalewski & Dvorak, *Top Stroke Rehabil*, 2011; Baert et al., *Disabil Rehabil*, 2012; Mansfield et al., *Stroke Res Treat*, 2016). The Discussion section should include some discussion of the fact that accelerometry alone might not be sufficient to capture physical activity post-stroke.

VERSION 1 – AUTHOR RESPONSE

Reviewer 1

Comment: Thank you for asking me to review this paper which I enjoyed reading very much. This was a very well thought through study resulting in a well written paper with comprehensive reporting

Response: Thank you

Comment: Patients recruited from one month post stroke – in reality few patients were recruited very early after stroke but some consideration for a future trial should be given if patients are still receiving ongoing rehabilitation such as ESD or community rehabilitation.

Response: To clarify, our inclusion criteria were to recruit people who were one month post discharge from NHS rehabilitation services and not one month post stroke. We have clarified this in the text by indicating that eligible participants could be taking part anytime post stroke (see page 5, first line of participants section).

It is however correct to say few patients were recruited very early after stroke (there was only one person within 3 months of stroke, see Table 1). We agree with the reviewer that consideration needs to be given on how we run our future trial and whether those still receiving ongoing rehabilitation should be included.

Comment: Adverse events were recorded and reported – it would be helpful to know what these consisted of - particularly adverse events at the venue.

We agree this detail is important and have now included a list of these events, see pages 17-18, footnotes to Tables 5a and 5b plus text on page 17 from 3rd line of Safety section.

Comment: Very many outcome measures applied which I imagine placed significant burden to patient and carer. For me one of the biggest limitations of the study was the lack of a robust ADL measure. It would have been extremely helpful if a measure of extended ADL had been recorded, eg the Nottingham Extended ADL. The study used the Patients Specific Functional Scale. This is a self-reported scale and not a commonly used scale in stroke rehabilitation research. It would have been hugely helpful if the authors had used an ADL measure that would allow comparison with other stroke rehab trials – for future meta-analyses.

Response: An a priori objective of the pilot was to evaluate the burden of measurement for participants. Interestingly most participants did not express any negative comments about the outcome measurement burden. Three carers did express concern about difficulty completing the Health Resource data section and the time this took. We have explained this carer concern on page 11, line 2 to 4 of sub-section b: Outcome Measure Burden.

We have added a statement in the limitations section (page 19, line 6-7 of limitations section) acknowledging the lack of robust ADL measures that is typically used in stroke research.

We suggest that the Patients Specific Functional Scale has some external validity beyond research for stroke survivors as it is used in many studies investigating long term conditions. That said we acknowledge it has limitations because of its individualised nature which limits robust between participant comparisons.

We are grateful for your suggestion to use the Nottingham Extended ADL measure (which we had considered using and will consider again) as well as your recommendation to use something that allows future meta-analyses. Thank you for this advice.

look forward to seeing this study progress to a large scale multi-centre study.

Thank you; we intend to submit a funding application for such a study in the very near future.

Reviewer 2

Comment: This article was extremely well written and of high quality. I recommend only a few minor revisions which the authors may like to consider.

Response: Thank you.

Comment: It appears ethics approval was given (found in the acknowledgements) but it is not written explicitly that approval was obtained.

Response: We confirm that ethics approval was given and we apologise for not making this clearer within the body of the manuscript. We have now moved our statement from the acknowledgement section into the main text, see page 5, lines 2-4 of Methods section.

Title

Comment: The title suggests it is the results of a pilot RCT yet the paper really is more than that and looks at feasibility and acceptability. Perhaps consider altering this to make it clearer to readers.

Response: Thank you we agree this could enhance the title of our manuscript and have now added 'investigating feasibility and acceptability' to the end of the title. See title page.

Comment: Abstract Line 16-20

eligibility should there be some note about having had a stroke?

Response: We agree, and this has been added – please see page 2, line 5 of the abstract.

Introduction

Comment: Page 4 line 7

UK spell out fully first time

Response: Corrected, see page 4, line 2 of the Introduction.

Methods

Comment: Page 6, lines 36-39

need semi-colons rather than commas to be consistent with above primary outcomes

Response: Corrected, see page 6 lines 14-17 of the Outcomes section.

Page 7

Comment: randomisation and blinding. Who actually created the randomisation? (It does say in the protocol, not sure if you want to add to this paper as well).

Response: We have clarified that the local CTU (Pen CTU) provided this service, see page 7, lines 4-5 of Randomisation and blinding section.

Comment: Was any computer package used to analyse the qual data? (eg NVivo). How many people analysed the qual data? How was content analysis conducted as well as the thematic analysis? Needs a little more detail. Economic evaluation would also benefit with more detail on how you came up with the results.

Response: We did not use NVivo for qualitative data management. One person (MN) led the qualitative analysis but this was then discussed (MN & SD), checked (RC) and agreed (MN, SD, LP, RC). Content analysis was performed to answer the pragmatic questions posed in the interviews (mostly about trial and intervention processes) and this is what is reported in this manuscript. A further, more in-depth, thematic analysis was also undertaken and will be reported in a companion paper currently in preparation. We were mindful of our word count and so did not provide this detail in the manuscript. However, we now have added a statement to cover the above points, see page 7, lines 7-10 of the Data Analysis section.

Likewise we have added a short statement providing more detail of the economic evaluation analysis process. Please see page 7 last 4 lines and page 8 first 3 lines.

Comment: Ethics approval did not seem clear in the paper but it was suggested in the acknowledgements, it would be better to be clearer on this.

Response: We confirm that ethics approval was given (REC ref: 15/SW/04) and we apologise for not making this clearer within the body of the manuscript. We have now moved our statement from the acknowledgement section into the main text; see page 5, lines 2-4 of Methods section.

Results page 8 Table 1

Comment: for gender remove the % for 70% or add it into 67. Include standard deviation after the mean age in years. Where there are mean values throughout the table I would include SD as you did in the MMSE.

Response: Thank you, we have added the '%' to 67 and included the SD after mean age in years, see Table 1, pages 8-9. There is only one other mean value in the Table and we have added the accompanying SD, please see Table 1, Pages 8-9.

Page 10 objective 3, line 40

Comment: The heading says acceptability and feasibility for participants and complete the Trainer manual, should the and be to complete? The and doesn't sound quite right

Response: Thank you, we have inserted the 'to'. See page 11, section heading Objective 3.

Page 10 line 55

Comment: How did the questionnaires place burden on the carer?

Response: We have clarified that it was the Health Resource Use questionnaire that was time consuming and hence a burden to some of the carers. See page 11, lines 2-4 of sub-section b Outcome Measure Burden.

Comment: Page 14 and 15, Table 4a and Table 4b. It would be good to have a few sentences actually describing these two tables.

Response: In line with our objective, we have added a couple of statements about these tables, please see end of page 14.

Page 16

Comment: the type of adverse or serious adverse events should be listed

Response: We agree this detail is important and have now included a list of these events. See pages 17-18, footnotes to Tables 5a and 5b plus text on page 17 from 3rd line of Safety section.

Discussion

Comment: Page 17, lines 15-17

would be good to actually state what some of the changes, refinements were rather than having to look through and try and work out which were changed etc.

Response: We have re-ordered some of our discussion points so that these changes are now described in the right place, see page 18, lines 13-17 of the Discussion section.

References number 15

Comment: add r to eport

Response: Corrected. See reference 15.

Reviewer 3

Major revision 1

The primary outcomes identified in the trial registration differ from those included in this paper, with the outcomes in the paper being focused on addressing feasibility objectives. While I agree that, as this is a pilot study, the outcomes should address feasibility issues, the authors must provide some justification for altering the plan outlined in the trial registration.

Response:

We recognise that trial registration details do not match our paper. The clinical trials website is not designed to accommodate pilot feasibility trials. We were unable to report our objectives as outcomes. Instead we were encouraged to report explicitly the outcome measures that participants would be required to complete. It is our understanding that this is to aid recruitment, to provide as full an explanation as possible as to what (and when) participants will need to complete measurements. Our plan was not altered - there was unfortunately this disconnect between what we could report as 'outcomes' on the trial registry and what our objectives were for the pilot feasibility trial. Although there is a disconnect it is important to note that the purpose of the study is outlined on the trial registry website under the 'detailed description' section (albeit brief) and there are links to the published protocol (where our objectives/outcomes are reported in full).

Major revision 2

There should be some hypotheses around the objectives identified in the last paragraph of Page 4. The authors conclude, in the 1st sentence of the Discussion, that the trial met its feasibility objectives. However, it is not clear how they arrived at this conclusion. For example, for objective 1, what recruitment numbers, randomisation rates etc would be acceptable to establish feasibility of a larger trial?

Response:

We politely decline to add specific hypotheses to our objectives as our study was not designed to undertake statistical testing. However, we have given more detail about how we made our assessment that our objectives had been met. Please see page 18, lines 3-6 of the Discussion section.

Major revision 3

More information about the experimental and control interventions are required. Consider the FITT principle (frequency, intensity, type, and time) when describing the exercise portion of the intervention; e.g., what specific exercises were included, how many repetitions/sets, how was intensity set (% of one-repetition maximum?), how was exercise dose progressed through the program? What was discussed in the home visits? For the control group, what was 'usual treatment'? These participants were discharged from the NHS rehabilitation services, so it seems that there was no treatment available to these participants; is this the case? Did the investigators document any treatment that control participants received outside of the trial (e.g., private rehabilitation services)?

Response:

Thank you for these important points about the reporting of our intervention and control conditions. We considered very carefully how much of the intervention programme was pre-specified in advance (for the trainers to use with participants) and also how much detail we were prepared to release in reporting our pilot trial results.

For the first point we worked with our trainers to decide how much detail and pre-specification (along the lines of FITT) was appropriate and they indicated that as every stroke survivor required an individually tailored programme it was not possible to pre-specify nor to uniformly prescribe the same exercise programme (in terms of FITT or progression) for all participants. Home visits were also individually tailored. It is therefore not possible to provide this level of detail here for 21 different participant programmes. Secondly we decided against publicising our trainer manual details or programme specification in any further detail here in the pilot trial (protocol or results manuscripts) as this would effectively reduce our ability to run an uncontaminated definitive trial.

We can confirm that 'usual treatment' is anything from nothing (most likely) through to any health services that the participant accesses. We requested that all trial participants did not participate in additional physical rehabilitation (either NHS or private) but we could not prevent them from doing so. We did not monitor control group participation in any treatments during the trial (as this contact would have been an additional intervention) but we did record health service use at the end of the trial for all participants. We have added a statement to this effect on page 6, lines 1-6 of the Control section.

Major revision 4

There is no statement regarding research ethics approval or participant consent in the paper.

Response: We confirm that ethics approval, including participant consent, was given and we apologise for not making this clearer within the body of the manuscript. We have now moved our statement from the acknowledgement section into the main text, please see page 5, lines 2-4 of Methods section.

Minor revision 6

Table 2: please include the MCIDs (or biologically/clinically meaningful effect sizes) and the standard deviation/variance used to calculate the sample sizes. Please clarify if this is the number of participants required per group, or the overall sample size for both groups combined. Why are ranges of sample sizes presented for some variables?

Response:

Thank you for this request as it has been a useful exercise to clarify what we have done. We have added the MCIDs (where known) and the SDs used to calculate the sample sizes and added a column to show effect sizes. We have clarified that sample size estimates are for both groups combined. We provide an explanation regarding why there are ranges and made sure an explanation is provided for each measure. Finally we have checked the calculations again to reflect that this conservative approach has been used consistently and corrected/added to the estimates column as appropriate (see Table 2 and footnotes, page 10-11).

Minor revision 7

Tables 4a and 4b: the captions state that means and standard deviations are presented, but there are three numbers in each cell; what is the third number? Also, please consider the precision with which numbers are presented; for example, was the Timed Up and Go measured with 10 ms precision?

Response: We have clarified that the third value represents the N (see Tables 4a and 4b, pages 15 and 16). We have indicated the precision of timing for the Timed Up and Go measure (it is to 10ms). The accelerometer measured movement 100 times a second (it was set to a sampling frequency of 100Hz); please see footnote to Table 4a, page 15.

Minor revision 8

Table 4b: include the name of the specific scales used to measure stroke & exercise self-efficacy, exercise beliefs, and quality of life.

Response: Where names exist we have added these, see Table 4b, page

Minor revision 9

Page 16, Line 7: replace "less" with "fewer".

Response: Corrected, see page 17, line 2.

Minor revision 10

'Safety' section: please provide further details regarding the nature of the adverse events (e.g., events possibly/probably/definitely related to the intervention might include muscle soreness or joint pain during/following exercise). Please clarify why some AEs are classified in Tables 5a and 5b as being 'related' when, in the text, it is stated that no AEs were definitely related to the intervention.

Response: We agree this detail is important and have now included a list of these events, see pages 17-18, footnotes to Tables 5a and 5b plus text on page 17 from 3rd line of Safety section. Thank you for spotting this error in our reporting. We have amended the text to clarify there were six serious adverse events and to detail what these were. Please see pages 17 and 18 as well as Tables 5a and 5b.

Minor revision 11

I do not understand the argument presented in the second paragraph of the Discussion regarding selection of a primary outcome measure for the larger trial. The investigators should select an outcome that measures what they would like to improve with the intervention (i.e., the target of the intervention). From the objectives presented on Page 5, the intervention aims to: enhance "function", develop self-management skills, and instill commitment to exercise. Therefore, the larger trial should assess if the intervention meets these objectives by including outcomes that measure "function", self-management skills, and adherence to exercise in daily life. Selecting outcome measures based *solely* on characteristics of outcomes measured during the pilot study (e.g., ceiling effects, high variability) does not make sense.

Response: We agree that the primary outcome measure should be selected on the basis that it assesses what we aimed to improve with the intervention. We selected our candidate primary outcome measures on this basis when planning this pilot trial. Thus the choice of which one to take forward to a definitive trial would then also be based on its psychometric utility. This is what we were considering in the discussion.

We have clarified this issue and added a brief statement to this effect at the start of the limitations section of the discussion, see page 19 lines 1-5 of the limitations section

Minor revision 12

Regarding measurement of physical activity, several investigators have suggested combining data from heart rate monitors, accelerometers, and physical activity questionnaires among individuals with stroke (e.g., see Resnick et al., *J Phys Act Health*, 2008; Zalewski & Dvorak, *Top Stroke Rehabil*, 2011; Baert et al., *Disabil Rehabil*, 2012; Mansfield et al., *Stroke Res Treat*, 2016). The Discussion section should include some discussion of the fact that accelerometry alone might not be sufficient to capture physical activity post-stroke.

Response:

Thank you for these useful resources and recommendations.

We agree that accelerometry is not a perfect measurement of PA quantification, nor are they specifically suited for all people with stroke. However in our participant population all our stroke survivors are able to stand and at least take a few steps. In addition our GeneActiv monitors are wrist worn, waterproof and with a battery that lasts for longer than the standard 7-day assessment period and so do not need to be removed once fitted – these features enhance compliance with wear. Furthermore the GeneActiv monitors are able to detect 'awake' time versus 'sleep' time and this differentiation provides useful objective PA versus sedentary behaviour data. These data will be reported in a companion paper currently in preparation.

Whilst we agree that the use of questionnaire assessment of PA alongside physiological measurement monitors may be useful in a future trial we also put forward a cautionary note: adding additional monitors, such as heart rate monitors, may risk participant burden and non-compliance...cont.

Minor revision 12 (cont)

As above

Response: Cont....We need to consider the feasibility and practicality of using additional physiological measurement in a much larger definitive trial but we agree that a psychometrically robust PA questionnaire may be better than the PA diaries we used in this pilot trial.

We have added a sentence to the discussion around the use of multiple PA measures. See page 19, lines 23-26 of the limitations section.

VERSION 2 – REVIEW

REVIEWER	Elissa Burton Curtin University, Australia
REVIEW RETURNED	28-Aug-2017

GENERAL COMMENTS	The authors have done a very good job in making the suggested changes. only one small change required in the title Objective 3 the second and needs to be removed.
--

REVIEWER	Avril Mansfield University Health Network, Canada
REVIEW RETURNED	31-Aug-2017

GENERAL COMMENTS	The majority of comments raised in my initial review have been addressed - thank you. However, I still believe the authors need to address my comment regarding hypotheses linked to the objectives. In this revised version, the authors have mentioned recruiting, attrition, randomization and retention targets. If these were a priori targets they must be mentioned in the Introduction section as 'hypotheses'.
--

VERSION 2 – AUTHOR RESPONSE

We trust this position is acceptable given the pilot nature of this work but thank our reviewer for raising the important issue of having clarity over research questions, objectives and hypotheses. We have however adjusted the text in the Background and Objectives section (page 4) to include our targets: 'Our objectives were to: 1) assess feasibility and acceptability of recruitment (target n=48), randomisation, allocation concealment and outcome assessment blinding; 2) determine retention rates (target of less than 20% attrition)'. We trust this is meets our reviewer's request to include this information in the Introduction.

VERSION 3 – REVIEW

REVIEWER	Elissa Burton Curtin University, Australia
REVIEW RETURNED	22-Sep-2017

GENERAL COMMENTS	The one minor adjustment I requested has been made. Thank you.
--